# Comparison of LC-MS^3^ and LC-MRM Method for Quantifying Voriconazole and Its Application in Therapeutic Drug Monitoring of Human Plasma

**DOI:** 10.3390/molecules27175609

**Published:** 2022-08-31

**Authors:** Wenbo Ren, Lei Yin, Gaixia Zhang, Taiyu Zhai, Jing Huang

**Affiliations:** 1Department of Laboratory Medicine, The First Hospital of Jilin University, Changchun 130061, China; 2School of Life and Pharmaceutical Sciences, Dalian University of Technology, Panjin 124221, China; 3Department of Laboratory Medicine, Nanfang Hospital, Southern Medical University, Guangzhou 510515, China

**Keywords:** HPLC-MS^3^, therapeutic drug monitoring, voriconazole

## Abstract

The TDM of voriconazole which exhibits wide inter-individual variability is indispensable for treatment in clinic. In this study, a method that high-performance liquid chromatography tandem mass spectrometry cubed (HPLC-MS^3^) is first built and validated to quantify voriconazole in human plasma. The system is composed of Shimadzu Exion LC^TM^ UPLC coupled with a Qtrap 5500 mass spectrometer. The separation of voriconazole is performed on a Poroshell 120 SB-C18 column at a flow rate of 0.8 mL/min remaining 7 min for each sample. The calibration curves are linear in the concentration range of 0.25–20 μg/mL. Intra-day and inter-day accuracies and precisions are within 8.0% at three concentrations, and the recoveries and matrix effect are all within accepted limits. In terms of stability, there is no significant degradation of voriconazole under various conditions. The HPLC-MS^3^ and HPLC-MRM (multiple reaction monitoring) methods are compared in 42 patients with Passing–Bablok regression and Bland–Altman plots, and the results show no significant difference between the two methods. However, HPLC-MS^3^ has a higher S/N (signal-to-noise ratio) and response than the MRM. Finally, the HPLC-MS^3^ assay is successfully applied to monitor the TDM (therapeutic drug monitoring) of voriconazole in human plasma, and this verifies that the dosing guidelines for voriconazole have been well implemented in the clinic and patients have received excellent treatment.

## 1. Introduction

Experts recommend voriconazole, which is the first available second-generation triazole drug, as primary therapy for invasive aspergillosis [1]. It is also used prophylactically by clinicians to avoid serious infections in immunosuppressed organ transplant recipients. The therapeutic drug monitoring (TDM) of voriconazole is not consistently recommended because it exhibits nonlinear pharmacokinetics, predictability of its accumulation or elimination is limited [2] and it has high interpatient and intrapatient variability. However, that the serum concentration of voriconazole is frequently monitored during and after severe inflammation is critical to maintain the serum concentration of voriconazole within the therapeutic range in elderly individuals [3]. Moreover, according to guideline recommendations, voriconazole serum concentrations should be monitored in patients with hepatic insufficiency, with drugs’ combination affecting voriconazole pharmacokinetics, with CYP2C19 gene mutations, with adverse drug events or poor efficacy of voriconazole, and which is life-threatening due to fungal infections. The TDM of voriconazole should also be determined in pediatric patients due to lack of experience with the drug and significant inter-individual variation [4]. Closely monitoring voriconazole pharmacokinetics also helps individualize antifungal therapy for children [5]. Thus, it is suggested that TDM of voriconazole may be indispensable for treatment.

For TDM of voriconazole, there are many analytical assays have been developed, such as immunoassays [6], high-performance liquid chromatography (HPLC) [7], gas chromatography mass spectrometry (GC-MS) [8] and liquid chromatography tandem mass spectrometry (LC-MS/MS) [9]. However, when a value close to the lower limit of the therapeutic range, an overestimation of immunoassay (systematic error of 0.39 μg/mL) is detected [10] and the enzyme multiplied immunoassay technique (EMIT)-measured levels are higher than those of HPLC [11]. Compared with HPLC assays with ultraviolet or fluorescence detection, the LC-MS/MS assays have improved selectivity, sensitivity, accuracy and precision. Just to our knowledge, it is not reported that the technique of MS^3^ is used to detect voriconazole in biological samples.

The MS^3^ detection, a scanning mode of Q-Q-Trap tandem mass spectrometry, for both of which the excitation efficiency and the scanning rate (20,000 Da/s) are significantly improved [12,13]. For MS^3^ detection, the precursor ions of analyte are firstly selected in Q1 and then fragmented in Q2 to produce ions that are captured in Q3. A specific product ion is then selected for secondary fragmentation in a linear ion trap, and second-generation fragment ions are scanned out to the detector. It improves the selectivity and lower limit of detection values by removing interference and background noise [14,15].

In this paper, we present an LC-MS^3^ method to intend to improve the quantitative comparison of voriconazole after medication of patients. As we know, this is the first time voriconazole has been quantified by using the LC-MS^3^ method in human plasma. Overall, there is a large intra- and inter-individual variability in voriconazole, and the need for individualized administration of voriconazole is well established.

## 2. Results and Discussion

### 2.1. Optimization of HPLC Conditions

A C18 column, which is suitable for the retention and separation of voriconazole from matrix components, is used for chromatography because it gives the best peak shape and resolution. Acetonitrile (ACN) is used as the organic solvent because of good retention and chromatographic resolution. Moreover, using 0.1% formic acid in water gives high signal intensity and negligible carryover. The rapid chromatographic separation and high sensitivity could be achieved by using a gradient elution method. Under the optimized conditions, the retention times of voriconazole and Carbamazepine-d2,15N (Car-d_2_,^15^N) are 4.14 and 3.89 min, respectively.

### 2.2. Optimization of MS Conditions

Both voriconazole and internal standard (IS) have the better sensitivity in positive ESI mode. The voriconazole and Car-d_2_,^15^N are detected by the transitions at *m*/*z* 350.3→224.3 and *m*/*z* 240.3→196.3 under MRM mode, respectively (Figure 1A,B). In the MS^3^ mode for voriconazole, the product ions at *m*/*z* 224.3 are further fragmented in the linear trap, and fragment ions at *m*/*z* 197.3 as a quantification trace and 126.9 are observed (Figure 1C). For IS, the daughter ions at m/z 181.3 are selected as a quantification trace (Figure 1D). Finally, the MS^3^ acquisitions use the transition ions at *m*/*z* 350.3→224.3→197.3 for detection of voriconazole (Figure 2A) and 240.3→196.3→181.3 for quantitation of IS (Figure 2B).

In the MS^3^, the collision energy (CE) is determined as 23 eV for voriconazole, 20 eV for IS. The excitation energy (AF_2_) is optimized to reveal a value of 0.1 V. The optimized MS parameters are shown in Table 1.

### 2.3. Sample Preparation

Based on the advantages of simplicity and rapidity, protein precipitation with methanol is selected for sample processing. In the research, the plasma: methanol 2:20 is found to give a high and stable recovery. To a 20 µL plasma is added 20 µL IS solution and 200 µL methanol to precipitate protein. Then, to ensure g enough sensitivity and ignored matrix effects, the supernatants from protein precipitation are diluted three times with water. In this study, the LOQ (lower limit of quantitation) of 0.25 μg/mL is sufficient and could be easily reduced by using more plasma or less dilution or more injection volume.

### 2.4. Assay Validation

As shown in representative LC-MS^3^ chromatograms of voriconazole and Car-d_2_,^15^N (Figure 3), the assay is free of significant interference at the retention times. The carry-over is negligible because of no enhancement for the response of voriconazole and Car-d_2_,^15^N in blank plasma samples (Figure 3A). In addition, cross-talk between MS channels is not observed at plasma samples (Figure 3C). The calibration curve (y = 0.1197x − 0.0181, r^2^ = 0.9996) shows good linearity in the range of 0.25–20 µg/mL. Accuracy and precision for the analysis of voriconazole in human plasma are shown in Table 2. Intra- and inter-day precision (relative standard deviation, RSD) are all within 8.72, and accuracies (relative error, RE) are all from 2.63 to 5.45 at the three concentrations. The actual concentrations as a percentage of nominal concentration for low, medium and high quality control (QC) samples, respectively, are shown by mean ± standard deviation (SD). Three concentrations of matrix effects are, respectively, as follows: 99.6 ± 4.5, 100.4 ± 5.6, 107.2 ± 2.8. No signification of suppression or enhancement signal is observed in both matrices. The recoveries are all within accepted limits (85~115%). The results also showed that the recoveries are all repeatable and concordant across the concentration range studied (Table 3). In terms of stability, the concentrations of voriconazole are within ±15% of nominal concentrations after storage at −80 °C for 2 weeks, 8 °C for 6 h, three freeze–thaw cycles and at room temperature (RT) for 8 h, which indicates there is no significant degradation of voriconazole under various storage conditions (Table 4).

### 2.5. Comparison of HPLC-MS^3^ and HPLC-MRM Methods

An LC-MRM method using transitions at *m*/*z* 350.3→224.3 for voriconazole and *m*/*z* 240.3→196.3 for Car-d_2_,^15^N is optimized and compared with the LC-MS^3^ method. Compared with MS^2^ acquisition, an MS^3^ scan could maintain a higher level of sensitivity. The peak height of voriconazole at 0.25 μg/mL is just 3397.0 cps, and S/N is 24.8 with the MS^2^ method (Figure 4A). However, for MS^3^ acquisition, the peak height is 7.3 × 10^5^ cps and S/N is 73.0 (Figure 4B). This is because the MS^3^ scan mode reduces matrix interference and background noise by adding a fragmentation step. Therefore, the MS^3^ has a higher S/N and response than the MRM.

### 2.6. The Novelty and Significance of the LC-MS^3^ Method

The MS^3^ technique is restricted to Qtrap MS systems and ion trap MS systems; the LC-MS/MS system comprises a HPLC with a QTRAP hybrid linear ion trap triple quadrupole mass spectrometer in this study. As our knowledge, this is the first report of the use of the LC-MS^3^ technique for quantification of voriconazole in human plasma and its application in therapeutic drug monitoring. The advantages of the LC-MS^3^ method include high selectivity, high sensitivity and high signal to noise ratio. Compared to GC-MS, the MS^3^ method includes high through-put (7 min per sample) and small sample volume (only 20 µL) [8]. Compared to the LC-MS/MS that was reported, protein precipitation with methanol was selected for sample processing in the MS^3^ method, which is simple and rapid [16]. This study offers a novel promising alternative technique to the traditional LC–MRM technique, benefiting from the high selectivity and high sensitivity of LC-MS^3^ Technique. 

### 2.7. Method Application

The validated HPLC-MS^3^ methods are applied for the quantification of the voriconazole concentration in 42 plasma samples from patients half an hour before drug administration. The comparison of the HPLC-MS^3^ and HPLC-MS^2^ method is shown in Figure 5. Passing–Bablok analysis to MS^2^ and MS^3^ methods provides a regression equation: y = −0.0187 + 1.001 x, p < 0.0001. The slope 95% CI included 1 (0.8830–1.1288), indicating that there is no proportional bias and 95% CI included 0 (−0.2213–0.1689), indicating the absence of an additional constant deviation (Figure 5A). A Bland–Altman test is also used to evaluate the agreement between the MRM and MS^3^ methods. In Figure 5B, that average concentration of voriconazole measured using two methods is as x-axis, and the percent difference of two methods as y-axis. In total, 95.2% of plots fell within maximum allowed difference (±1.96 SD). These results suggest that two methods can be reliably exchanged in analysis of voriconazole in human plasma.

The valley concentration of voriconazole is illustrated using HPLC-MS^3^ in Figure 6 and Appendix A. The result shows that these expected values from 5.52 μg/mL are within the reference range from 0.50 μg/mL to 5.00 μg/mL [4]. This verified that the dosing guidelines for voriconazole have been well implemented in the clinic and patients have received excellent treatment.

## 3. Materials and Methods

### 3.1. Reagents and Chemicals

Standards for voriconazole (Figure 7A) were purchased from the A Chemtek Inc (ACT). Car-d_2_,^15^N for using as IS (Figure 7B) was provided by United States Biological. HPLC grade ACN and methanol were purchased from Fisher (Fair Lawn, NJ, USA). The formic acid was purchased from Merck (Darmstadt, Germany). Ultra-high purity water was prepared using a Milli-Q System (Millipore, Bedford, MA, USA).

### 3.2. LC-MS^3^ Conditions

Chromatography was performed on a Shimadzu Exion LC^TM^ UPLC system (Kyoto, Japan) equipped with a binary pump, a degasser, a thermostatically controlled column compartment set at 40 °C and an auto sample manager (Kyoto, Japan) maintained at 8 °C. Separation was performed on a Poroshell 120 SB-C18 column (4.6 × 50 mm, 2.7 μm) using gradient elution with 0.1% formic acid in water (solvent A) and acetonitrile (solvent B) at a flow rate of 0.8 mL/min. The separation gradient program is as follows: 0.0–2.0 min, (25.0% B); 2.0–3.0 min, (25.0–65.0% B); 3.0–3.5 min: (65–75.0% B); 3.5–4.0 min (75.0–90.0% B); 4.0–4.5 min (90.0% B); 4.5–4.6 min (90.0–25.0% B); 4.6–7.0 min: (25.0% B).

MS analysis employed a Qtrap 5500 mass spectrometer (AB Sciex, Foster City, Canada) equipped with a TurboIonSpray™ source operated in the positive ion mode. In the MS^3^ mode, first generation product ions (MS^2^) were fragmented to yield the second generation product ions (MS^3^), which then underwent MS^3^ transitions. MS conditions were optimized by the syringe pump infusion of standard solutions of voriconazole and Carbamazepine-d2,15N. The optimized parameters were shown in Table 1. Data acquisition was controlled by Analysis 1.6.3 software.

### 3.3. Preparation of Calibration Standards and Quality Control Samples

A stock solution of voriconazole was prepared in methanol:water (50:50). Calibration standards were prepared by diluting standard solutions with blank human plasma to a final concentration of 0.25, 0.5, 1.0, 2.5, 5.0, 10.0 and 20.0 μg/mL. QC samples of 0.5, 2.5 and 10.0 μg/mL were prepared independently in a similar manner. A IS stock solution was diluted to 5 μg/mL. All solutions were stored at −80 °C until use.

The 20 μL calibration standards and QC samples were added to 20 μL IS working solution and 280 μL methanol. After vortexing for 5 min and centrifuging at 4 °C and at 15,000 rpm for 10 min, the mixture was diluted three times with water. Then, 2 μL of the supernatant was injected into the LC-MS system for analysis.

### 3.4. Assay Validation

The assay validation was performed in accordance with the biological method validation guidance of the U.S. Food and Drug Administration (FDA) [16,17]. The procedures for assay validation were selectivity, linearity, LOQ, extraction recovery, matrix effect, accuracy, precision and stability. The details are shown in Appendix A.

### 3.5. Clinical Application

In order to demonstrate the applicability of the HPLC–MS^3^ method, 42 plasma samples from intensive care unit patients under treatment obtained from the First Hospital of Jilin University in 2021 were analyzed. The analysis of these samples was conducted to quantify the plasma levels of voriconazole. All plasma samples were obtained after centrifugation of blood in K_2_ EDTA vacutainer tubes which were immediately stored frozen (–20 °C) until sample preparation. The study is approved by the Human Research Ethics Committee of the First Hospital, Jilin University. Written informed consent was obtained from all subjects. The HPLC–MS^3^ method was compared to the HPLC–MS^2^ method to measure the concentrations of voriconazole from patients.

For analysis, 20 μL human plasma was added 20 μL IS working solution and 280 μL methanol, then vortexed for 5 min and centrifuged at 15,000 rpm and at 4 °C for 10 min. Then, after three times dilution, the 5 μL of the supernatant was injected into the LC-MS.

### 3.6. Statistical Analysis

Microsoft Excel 2016 and MedCalc were used to carry out data processing and graphic presentation. Passing–Bablok regression and Bland–Altman plot analyses were applied to evaluate the agreement between the HPLC-MRM and HPLC-MS^3^ method. Only when more than 67% of the sample pairs deviate within 1.96SD of the mean difference was the method sufficient to be an alternative quantitative method [17,18].

## 4. Conclusions

In this study, a simple, selective and high throughput LC-MS^3^ method for quantification of voriconazole in human plasma is developed and validated. The application of this LC-MS^3^ assay is completed on clinical samples, and it is proved that the developed LC–MS^3^ method is accurate and reliable. This work verifies that the dosing guidelines for voriconazole have been well implemented in the clinic and patients have received excellent treatment.

## Figures and Tables

**Figure 1 molecules-27-05609-f001:**
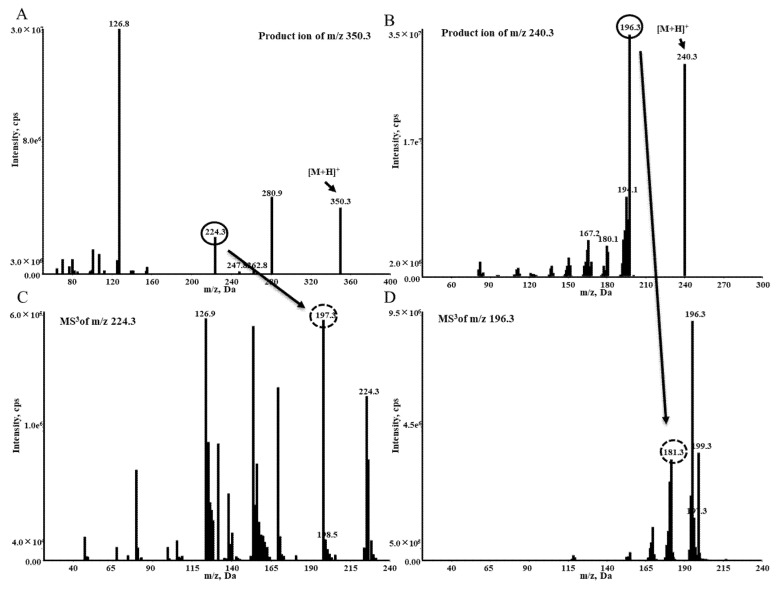
Representative product ion (MS^2^) for (**A**) voriconazole and (**B**) carbamazepine-d2,15N and MS^3^ for (**C**) voriconazole and (**D**) carbamazepine-d2,15N.

**Figure 2 molecules-27-05609-f002:**
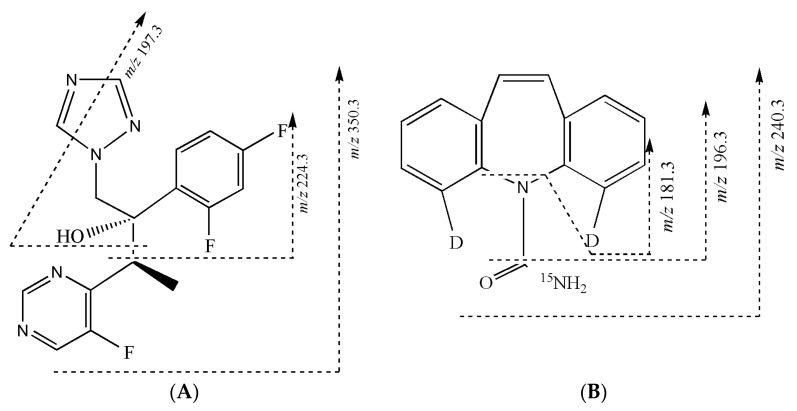
The fragmentation of (**A**) voriconazole and (**B**) carbamazepine-d2,15N.

**Figure 3 molecules-27-05609-f003:**
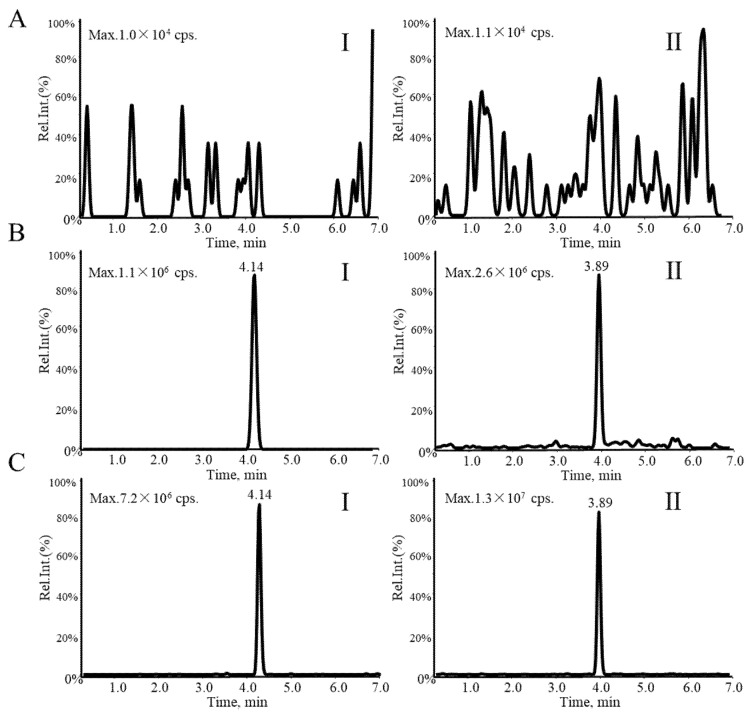
LC-MS^3^ chromatogram of voriconazole (I) and carbamazepine-d2,15N (II) in (**A**) blank plasma, (**B**) at the LOQ with 0.25 μg/mL voriconazole and 5 μg/mL IS, and (**C**) a plasma sample.

**Figure 4 molecules-27-05609-f004:**
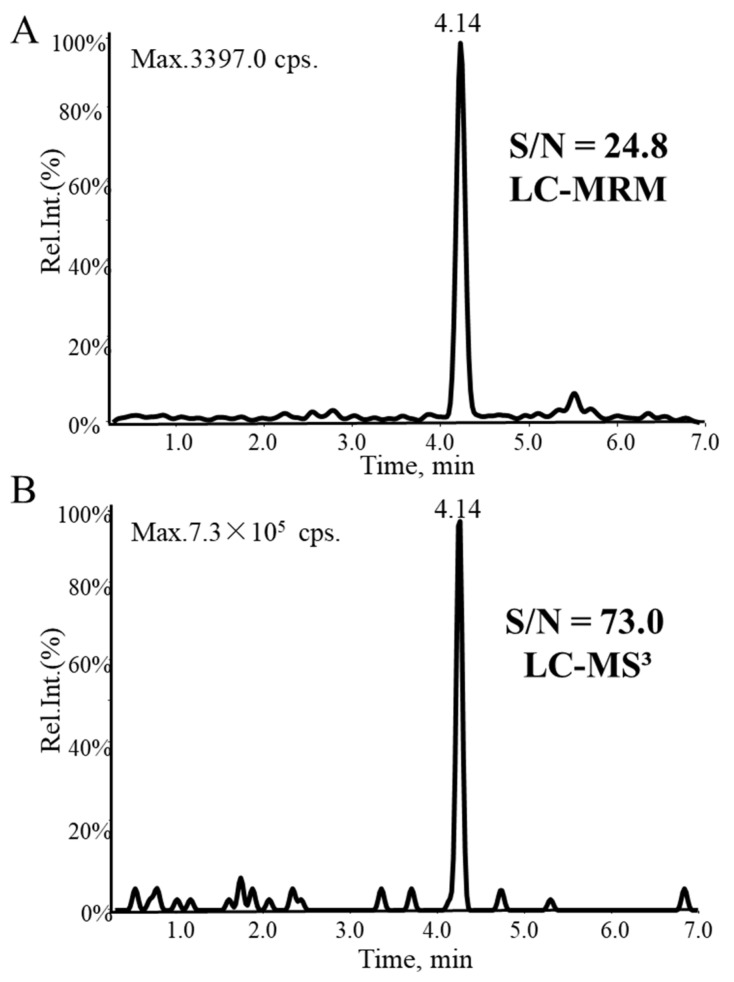
Representative chromatograms of voriconazole at the LOQ (0.25 μg/mL) analyzed by (**A**) LC-MRM and (**B**) LC-MS^3^. The S/N values are reported on the top of each peak.

**Figure 5 molecules-27-05609-f005:**
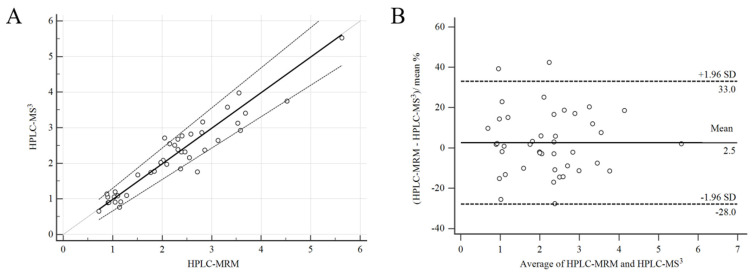
Comparison of voriconazole concentration in patient samples using HPLC-MRM and HPLC-MS^3^. (**A**) The solid black lines are the Passing–Bablok regression. (**B**) Bland–Altman analysis verifies the difference of voriconazole concentration measured using HPLC-MS^2^ and HPLC-MS^3^ in 42 human plasma samples.

**Figure 6 molecules-27-05609-f006:**
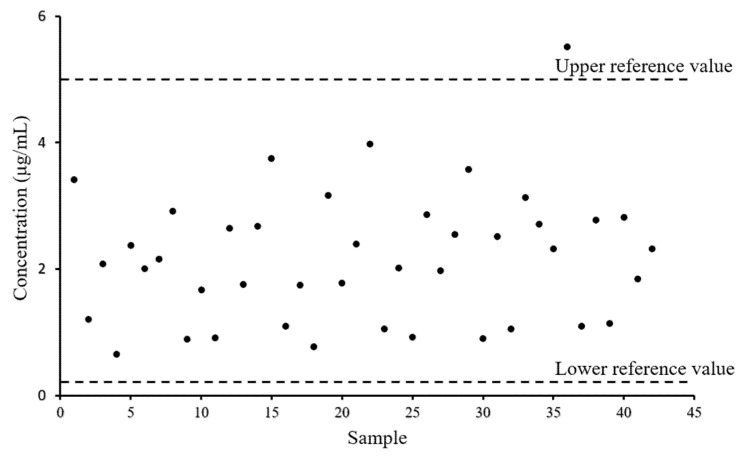
The concentration of voriconazole in 42 plasma samples from patients half an hour before drug administration using HPLC-MS^3^.

**Figure 7 molecules-27-05609-f007:**
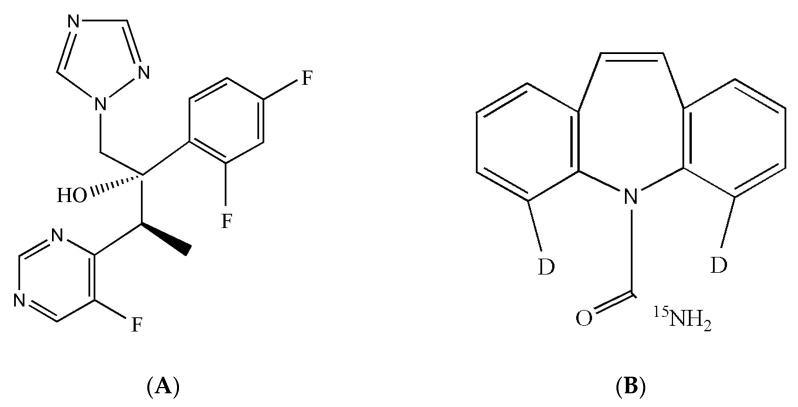
Structures of (**A**) voriconazole and (**B**) Carbamazepine-d2,15N.

**Table 1 molecules-27-05609-t001:** Optimized parameters for quantitation of voriconazole using LC-MS^3^.

Parameters	MS^3^	
	Voriconazole	IS
MS^3^ transitions	350.3→224.3→197.3	240.3→196.3→181.3
Declustering potential (V)	80	80
Entrance potential (V)	10	10
Collision energy (eV)	23	20
Excitation energy (AF2) (V)	0.1	0.1
Scan rate (Da/s)	10,000	10,000
LIT fill time (ms)	80	80
Excitation time (ms)	25	25
Turboheater temperature (°)	450	450
Ionspray voltage (V)	5500	5500
Curtain gas (N_2_, psi)	30	30
Nebulizer gas (N_2_, psi)	50	50
Heater gas (N_2_, psi)	50	50

**Table 2 molecules-27-05609-t002:** Inter- and intra-day precision and accuracy for voriconazole in human serum (data are based on assay of six replicates on three different days).

Drug	QC (μg/mL)	Intra-Day Precision (RSD%)	Inter-Day Precision (RSD%)	Accuracy (RE%)
	0.5	8.72	0.52	2.63
Voriconazole	2.5	6.64	2.37	4.96
	10	3.66	3.68	5.45

**Table 3 molecules-27-05609-t003:** Absolute matrix effects (%) and recoveries (%) for voriconazole (data are mean ± SD for *n* = 4).

Drug	QC	Matrix Effects (%)	Recovery (%)
Voriconazole	0.5 2.5 10	99.6 ± 4.5 100.4 ± 5.6 107.2 ± 2.8	107.2 ± 7.7 103.6 ± 5.7 96.1 ± 6.0

**Table 4 molecules-27-05609-t004:** Stability of voriconazole under various storage conditions (data are mean ± SD, *n* = 3).

	Low QC	Middle QC	High QC
Long-term (−80 °C)	98.5 ± 12.1	101.7 ± 5.8	98.2 ± 5.1
Three Freeze-thaw	100.7 ± 4.2	108.0 ± 1.7	95.1 ± 5.6
Under autosampler conditions (8 °C)	95.3 ± 6.7	101.9 ± 9.0	101.9 ± 3.2
Short-terms (4 h, RT)	97.2 ± 5.3	102.1 ± 4.0	102.9 ± 3.3

Abbreviation: RT, room temperature.

## Data Availability

The data presented in this study are available on request from the corresponding author.

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
