# Peer review of "Comparison of LC-MS3 and LC-MRM Method for Quantifying Voriconazole and Its Application in Therapeutic Drug Monitoring of Human Plasma"

_molecules, 2022, doi:10.3390/molecules27175609_

Round 1
Reviewer 1 Report
Despite the fact that the study covered all aspects of analytical chemistry protocols in a nice way, there are many points to consider that will strengthen the study.
1- The second paragraph of the introduction section should be moved to the first paragraph.
2- Chemdraw should be used to redraw the chemical structure in Fig.1; it cannot be adapted in this style.
3- In figure 2, I recommend drawing the fragmentation proposal in the different figure.
4- The authors need to present the fragmentation of Voriconalzole and IS in separate figures, explaining why they believe that to be the correct fragmentation. They can add it to section 2.2.
5- Why carbamazipine was chosen for IS?
6- LLOQ and LLOD should be changed to LOQ and LOD in the entire study. 7- The effects of the matrix should be explained.
8- (A, B, C) should be stated clearly in Figure 3 as in its legend.
9- "Compared with HPLC assays with ultraviolet or fluorescence detection, the LC-MS/MS assays have improved selectivity, sensitivity, accuracy and precision. Just to our knowledge, it was not reported that the technique of MS3 was used to detect voriconazole in biological samples". If you could briefly discuss your results compared to these methods again in the discussion section and explain why you think your method is superior?
Author Response
Despite the fact that the study covered all aspects of analytical chemistry protocols in a nice way, there are many points to consider that will strengthen the study.
The main corrections in the paper and the responses to your comments are as flowing:
1- The second paragraph of the introduction section should be moved to the first paragraph.
Response: Thank you for your comment. We have made the correction according to your comment.
2- Chemdraw should be used to redraw the chemical structure in Fig.1; it cannot be adapted in this style.
Response: Thank you for your comment. We have redrawn the chemical structure according to your comment.
(Please see the attachment because the picture cannot be opened)
3- In figure 2, I recommend drawing the fragmentation proposal in the different figure.
Response: Thank you for your comment. We have added the fragmentation of voriconazole and carbamazepine-d2,15N in Figure 3 according to your comment. The added images were shown below.
(Please see the attachment because the picture cannot be opened)
Figure 3. The fragmentation of of (A) voriconazole and (B) carbamazepine-d2,15N.
4- The authors need to present the fragmentation of Voriconalzole and IS in separate figures, explaining why they believe that to be the correct fragmentation. They can add it to section 2.2.
Response: Thank you for your comment. We have added the fragmentation of voriconazole and carbamazepine-d2,15N in Figure 3 and explanation of correct fragmentation to section 2.2 according to your comment. The added images and e explanation were shown below.
(Please see the attachment because the picture cannot be opened)
Figure 3. The fragmentation of of (A) voriconazole and (B) carbamazepine-d2,15N.
Supplement of section 2.2:
In the MS3 mode for voriconazole, the product ions at m/z 224.3 are further fragmented in the linear trap and fragment ions at m/z 197.3 as a quantification trace and 126.9 were observed. For IS, the daughter ions at m/z 181.3 was selected for as a quantification trace.
5- Why carbamazipine was chosen for IS?
Response: Thank you for your comment. Carbamazepine was chosen as I.S. due to the similarity of its extraction efficiency, matrix effects and retention behaviour with voriconazole. The LC-MS3 chromatogram of voriconazole (â… ) and carbamazepine-d2,15N (â…¡) at the LOQ with 0.25 μg/mL voriconazole and 5 μg/mL IS in the following figure.
(Please see the attachment because the picture cannot be opened)
6- LLOQ and LLOD should be changed to LOQ and LOD in the entire study.
Response: Thank you for your comment. We have made the correction in paper according to your comment.
7- The effects of the matrix should be explained.
Response: Thank you for your comment. We have added the explain as following:
Three concentrations of matrix effects respectively were as follows: 99.6 ± 4.5, 100.4 ± 5.6, 107.2 ± 2.8. The matrix effects which were all within accepted limits (85%~115%) revealed no significant ion suppression or enhancement signal was observed in both matrices. The validation details of the matrix effects were described in Supplement material.
8- (A, B, C) should be stated clearly in Figure 3 as in its legend.
Response: Thank you for your comment. We have re-write this part according your comment as following: As shown in representative LC-MS3 chromatograms of voriconazole and Car-d2,15N (Figure 4), the assay was free of significant interference at the retention times. The carry-over was negligible because of no enhancement for the response of Voriconazole and Car-d2,15N in blank plasma samples (Figure 4A). In addition, cross-talk between MS channels was not observed at plasma samples (Figure 4C). (Because of the addition of a figure, Figure 3 becomes Figure 4).
9- "Compared with HPLC assays with ultraviolet or fluorescence detection, the LC-MS/MS assays have improved selectivity, sensitivity, accuracy and precision. Just to our knowledge, it was not reported that the technique of MS3 was used to detect voriconazole in biological samples". If you could briefly discuss your results compared to these methods again in the discussion section and explain why you think your method is superior?
Response: Thank you for your comment. The comment was valuable and very helpful for improving our paper. We added the section 2.6 in paper.
2.6 The novelty and significance of the LC-MS3 method
MS3 technique is restricted to Qtrap MS systems and ion trap MS systems, the LC-MS/MS system comprised a HPLC with a QTRAP hybrid linear ion trap triple quad-rupole mass spectrometer in this study. As our knowledge, this is the first report of the use of LC-MS3 technique for quantification of voriconazole in human plasma and its application to a therapeutic drug monitoring. The advantages of this developed LC-MS3 method include high selectivity, high sensitivity, high signal to noise ratio. Compared to GC-MS, MS3 method include high through-put (7 minutes per sample), and small sample volume (only 20µL) [16]. Compared to the LC-MS/MS had reported, Acetonitrile (ACN) was used as protein precipitation in MS3 method, which was simple and quick [17]. This study offers a novel promising alternative technique to traditional LC–MRM technique owing to the high selectivity and high sensitivity of LC-MS3 Technique.
Special thanks to you for your good comments.

Reviewer 2 Report
This manuscript compared two analytical platforms (LC-MS3 and LC-MRM) to quantify the concentration of voriconazole in human plasma. Monitoring the drug concentration in the human body is critical. This research provides an efficient way to measure the target compounds rapidly and accurately. These analytical tools make personalized administration and risk regulations possible and feasible. The reviewer considers this manuscript can be accepted by Molecules, but several minor flaws should be improved:
1. When the abbreviations appear on the paper for the first time, the full name should be mentioned, such as MRM, IS, CI, etc.
2. In Figure 1, voriconazole is a chiral compound. Its stereo configuration has been determined. Therefore, the structure should exhibit its stereo information.
3. Figure 3 is not clear. A), B), and C) were not shown in the figure.
4. How was the LLOQ calculated? 0.25 ug/mL was the lowest concentration that can be prepared in this study, or was it calculated by the calibration curve and s/n? What is the LLOQ of the method in MRM?
5. The language requires further polishing. Several typos and grammar mistakes need the authors to go through carefully.
Author Response
This manuscript compared two analytical platforms (LC-MS3 and LC-MRM) to quantify the concentration of voriconazole in human plasma. Monitoring the drug concentration in the human body is critical. This research provides an efficient way to measure the target compounds rapidly and accurately. These analytical tools make personalized administration and risk regulations possible and feasible. The reviewer considers this manuscript can be accepted by Molecules, but several minor flaws should be improved:
The main corrections in the paper and the responses to your comments are as flowing:
1. When the abbreviations appear on the paper for the first time, the full name should be mentioned, such as MRM, IS, CI, etc.
Response: Thank you for your comment. We have corrected the abbreviations according to your comment.
2. In Figure 1, voriconazole is a chiral compound. Its stereo configuration has been determined. Therefore, the structure should exhibit its stereo information.
Response: Thank you for your comment. We have redrawn the stereo configuration of voriconazole in Figure 1A.
(Please see the attachment because the picture cannot be opened)
3. Figure 3 is not clear. A), B), and C) were not shown in the figure.
Response: Thank you for your comment. We have modified the Figure and the legend according to your comment. In figure 3, the A, B and C, and â… and â…¡ were shown.
4. How was the LLOQ calculated? 0.25 ug/mL was the lowest concentration that can be prepared in this study, or was it calculated by the calibration curve and s/n? What is the LLOQ of the method in MRM?
Response: Thank you for your comment.
Question 1: LLOQ is the lowest concentration of the standard curve that can be measured with acceptable accuracy and precision.
Question 2: The S/N of LLOQ should be larger than 10.
Question 3: The LLOQ of the method in MRM also was 0.25 ug/mL. The same LLOQ is used for MRM and MS3 in order to compare the signal-to-noise ratio and signal response strength at the same level.
5. The language requires further polishing. Several typos and grammar mistakes need the authors to go through carefully.
Response: Thank you for your comment. The language has further polished in paper.
Special thanks to you for your good comments.

Reviewer 3 Report
The manuscript contains a study on the comparison of methods for the quantification of voriconazole, which is used as a drug in antifungal therapy in children. The subject is very important due to the possibility of practical application of the method, however, the presented results have an average level of innovation. The manuscript requires major revisions and editorial corrections before it is ready for publication.
Further detailed comments for consideration are provided below.
Comment 1#
The results obtained were not discussed with other works in which voriconazole and also other antifungal drugs were determined in human plasma. Complete the discussion.
Comment 2#
The author uses the name „Voriconazole” with a capital letter, although this is not the beginning of a sentence. Correct this error in all places.
Comment 3#
Correct other editing errors such as missing spaces (mostly on quotes). Due to the lack of line numbers, I am unable to provide specific locations.
Author Response
The manuscript contains a study on the comparison of methods for the quantification of voriconazole, which is used as a drug in antifungal therapy in children. The subject is very important due to the possibility of practical application of the method, however, the presented results have an average level of innovation. The manuscript requires major revisions and editorial corrections before it is ready for publication.
The main corrections in the paper and the responses to your comments are as flowing:
Comment 1#
The results obtained were not discussed with other works in which voriconazole and also other antifungal drugs were determined in human plasma. Complete the discussion.
Response: Thank you for your comment. The comment is valuable and very helpful for improving our paper. We added the discussion section 2.6 in paper.
2.6 The novelty and significance of the LC-MS3 method
MS3 technique is restricted to Qtrap MS systems and ion trap MS systems, the LC-MS/MS system comprised a HPLC with a QTRAP hybrid linear ion trap triple quad-rupole mass spectrometer in this study. As our knowledge, this is the first report of the use of LC-MS3 technique for quantification of voriconazole in human plasma and its application in therapeutic drug monitoring. The advantages of this developed LC-MS3 method include high selectivity, high sensitivity, high signal to noise ratio. Compared to GC-MS, MS3 method include high through-put (7 minutes per sample), and small sample volume (only 20µL) [16]. Compared to the LC-MS/MS had reported, protein precipitation with methanol is selected for sample processing in MS3 method, which is simple and rapid [17]. This study offers a novel promising alternative technique to traditional LC–MRM technique benefiting from the high selectivity and high sensitivity of LC-MS3 Technique.
Comment 2#
The author uses the name “Voriconazole” with a capital letter, although this is not the beginning of a sentence. Correct this error in all places.
Response: Thank you for your comment. We have changed “Voriconazole” to “voriconazole” except the beginning of a sentence.
Comment 3#
Correct other editing errors such as missing spaces (mostly on quotes). Due to the lack of line numbers, I am unable to provide specific locations.
Response: Thank you for your comment. We have corrected all editing errors and added line numbers according to your comment.
Special thanks to you for your good comments.

Round 2
Reviewer 1 Report
I have no further comments
Reviewer 3 Report
Good luck with your future scientific work.